Corrected: Publisher correction

# High power surface emitting terahertz laser with hybrid second- and fourth-order Bragg gratings

Yuan Jin[1], Liang Gao[1], Ji Chen[1], Chongzhao Wu[1], John L. Reno[2] & Sushil Kumar[1]

A surface-emitting distributed feedback (DFB) laser with second-order gratings typically excites an antisymmetric mode that has low radiative efficiency and a double-lobed far-field beam. The radiative efficiency could be increased by using curved and chirped gratings for infrared diode lasers, plasmon-assisted mode selection for mid-infrared quantum cascade lasers (QCLs), and graded photonic structures for terahertz QCLs. Here, we demonstrate a new hybrid grating scheme that uses a superposition of second and fourth-order Bragg gratings that excite a symmetric mode with much greater radiative efficiency. The scheme is implemented for terahertz QCLs with metallic waveguides. Peak power output of 170 mW with a slope-efficiency of 993 mW A$^{-1}$ is detected with robust single-mode single-lobed emission for a 3.4 THz QCL operating at 62 K. The hybrid grating scheme is arguably simpler to implement than aforementioned DFB schemes and could be used to increase power output for surface-emitting DFB lasers at any wavelength.

[1] Department of Electrical and Computer Engineering, Lehigh University, Bethlehem, PA 18015, USA. [2] Sandia National Laboratories, Center of Integrated Nanotechnologies, MS 1303, Albuquerque, NM 87185, USA. Correspondence and requests for materials should be addressed to Y.J. (email: yuj314@lehigh.edu) or to S.K. (email: sushil@lehigh.edu)

High-power surface-emitting (SE) semiconductor lasers[1, 2] have significant advantages over their edge-emitting counterparts in multiple aspects related to coupling and alignment optics, testing and packaging, power scaling through arrays, wavelength stability, immunity to facet damage among others. The radiative efficiency of SE lasers has been increased by using different techniques, including use of curved and chirped gratings for infrared diode lasers[3, 4], implementation of central grating π shift for short-cavity devices[5], plasmon-assisted mode selection for mid-infrared quantum cascade lasers (QCLs)[6–8], and graded photonic structures for terahertz QCLs[9]. At near-infrared wavelengths, such high-power lasers are predominantly realized as vertical-cavity SE lasers (VCSELs)[10]. However, some of the highest power output for single-mode lasers has been demonstrated with second-order DFB gratings[3, 4]. For QCLs that emit at longer wavelengths[11–13], vertical cavities are not possible owing to the transverse-magnetic polarization of the intersubband optical field, hence second-order gratings are used for surface-emission[7, 8, 14–18]. High-brightness single-mode QCLs are desired across the mid-IR and terahertz spectrum for applications in chemical and biomolecular sensing and spectroscopy. At mid-IR wavelengths, multi-Watt level power was demonstrated for edge-emitting DFB QCLs[19]; however, for surface (or substrate) emitting DFB QCLs with second-order gratings, the power output is yet to reach such levels[7, 8, 20] and there is considerable scope for improvement. For terahertz QCLs, the sub-wavelength confinement of the metallic cavities[11] offered new challenges to development of DFB techniques. Consequently, a variety of DFB configurations have been demonstrated[21, 22]. A modification of the one-dimensional second-order gratings with graded periodicity led to the previous highest power output of 103 mW at 20 K for single-mode terahertz QCLs[9]. More recently, similar power levels have been realized with external-cavity SE terahertz QCLs[23], which offer the advantage of low-divergence beams at the cost of frequency-stability and specificity afforded by DFB lasers.

Here we describe a new scheme for enhancing radiative efficiency for surface-emitting DFB QCLs in metallic cavities that achieves record high output power for single-mode terahertz QCLs. A record high slope-efficiency is realized that is more than four times greater than that in ref.[9], and is also considerably higher than that from terahertz QCLs with single-plasmon waveguides that have recently reached Watt level output powers[24, 25].

## Results

**Concept**. A periodic perturbation in an optical waveguide leads to Bragg diffraction up to multiple higher orders that could be used to couple counter-propagating waves in the waveguide to establish DFB. The following equation describes the momentum conservation relation between the wavevectors of the incident guided wave inside the cavity $k_i \approx 2\pi/\lambda_{wg} = 2\pi n_{eff}/\lambda$ (where $\lambda_{wg}$ is the wavelength inside the waveguide, $\lambda$ is the free space wavelength, and $n_{eff}$ is the effective index of propagation) and that of the diffracted wave $k_d$, which could be outside or inside the cavity at any angle $\theta_d$ (as defined with respect to the surface-normal). This is also represented schematically in Fig. 1a.

$$n\frac{2\pi}{\Lambda} = k_i + k_d\sin(\theta_d) \qquad (1)$$

here $\Lambda$ is the grating period, $2\pi/\Lambda$ is the grating wavevector, and $n$ is an integer ($n = 1,2,3 \ldots$) that specifies the diffraction order. From this equation, it can be concluded that a $n$-th order grating structure, where $n$ is an even number, causes $n/2$-th order diffraction to occur in the surface-normal direction.

The concept of the hybrid second-order and fourth-order DFB gratings for increased radiative efficiency of SE lasers is described next. Figure 1b shows the conventional second-order gratings in an optical cavity that are used to realize broad-area surface (or substrate) normal emission from the cavity, since the first-order Bragg diffraction is perpendicular to the guided wave direction in the cavity as per Eq. (1)[1]. Coupled-mode[2] or finite-element modeling of the periodic structures are used to compute the photonic bandstructure. Typically, the band-edge modes in a longitudinal cavity are termed as symmetric and antisymmetric with high and low radiative efficiencies respectively, where the latter is excited for a SE DFB laser thereby limiting its output power. The radiative efficiency depends on the precise amplitude and phase of the standing-wave (resonant-mode) with respect to the periodicity of the grating. This behavior is typical of different types waveguides and DFB gratings and not just metallic cavities that are shown in Fig. 1b. Figure 1c shows conceptually how superposing a fourth-order grating structure on the existing second-order grating serves to enhance the overall radiative efficiency. Due to weaker Bragg diffraction for feedback, the fourth-order grating does not alter the modeshapes of band-edge modes significantly, which changes the electromagnetic field distribution along the length of cavity only slightly, that is, the maximum point of $E_x$ would slight offset the slit center and reduce the radiative efficiency of symmetric mode. An array of antenna-like dipoles with opposite polarities are introduced in each repeat period of the hybrid grating that causes destructive interference for the outcoupled radiation (marked as competing dipole from 4th order DFB as shown in Fig. 1c), which in turn reduces the out-coupling efficiency of symmetric mode. However, since the second-order Bragg diffraction is in the surface-normal direction, the fourth-order grating could be located so as to increase radiative loss for the original antisymmetric mode of the second-order DFB structure. Such a hybrid grating enhances the radiative efficiency of the antisymmetric mode and reduces that of the symmetric mode, and hence, either of those band-edge modes could possibly be excited as per the precise implementation of the fourth-order grating. In either case, the radiative efficiency is enhanced when compared to the excited mode for the original second-order DFB structure. It is to be noted that the 1st and 3rd order diffraction from the fourth-order gratings DFB could potentially produce off-normal-radiation since they are not diffracted perpendicularly or parallel to the surface. However, the possibility of such radiation does not exist in such a cavity, which can be shown through straightforward analysis based on Eq. (1). Finite-element-simulations were also carried out to validate this claim, which is described further in the Supplementary Note 6.

**Implementing hybrid DFB scheme for terahertz QCLs**. To describe the specific implementation of the hybrid DFB scheme for terahertz QCLs with metallic cavities, and a comparison with the conventional second-order DFB in such QCLs[18], results from finite-element simulations are shown in Fig. 2. The radiative surface losses of the resonant modes, their frequencies, and electric field profiles for the band-edge modes for finite-length cavities with gratings implemented in the top metal cladding are shown. It is argued that the value of radiative loss itself is a direct indicator of the out-coupling efficiency from such DFB cavities. Supplementary Note 7 further elaborates this aspect about the out-coupling efficiency from the cavity. For a conventional second-order DFB, the radiative loss for the upper band-edge mode (symmetric in-plane ($E_x$) field) is considerably larger than the non-radiative lower band-edge mode (antisymmetric in-plane

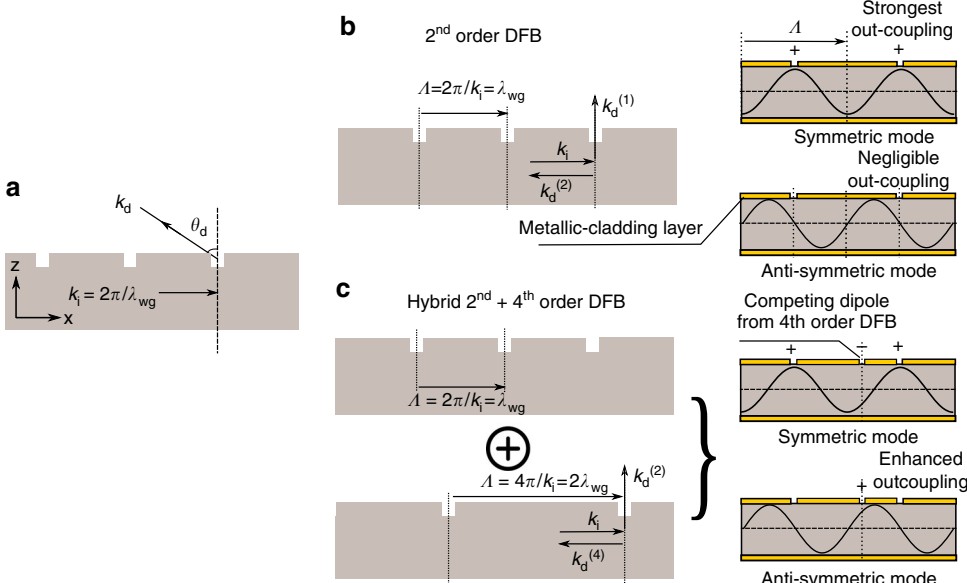

**Fig. 1** The hybrid second-order and fourth-order grating scheme. **a** Schematic depicting Bragg diffraction due to periodic gratings in an optical waveguide. $k_i$ is the incident wavevector of the guided wave with a wavelength $\lambda_{wg}$ and $k_d$ is the wavevector of a diffracted wave, $\theta_d$ is the angle of diffracted wave. **b** DFB due to second-order gratings, in which first-order Bragg diffraction $(k_d^{(1)})$ is in the surface-normal direction and second-order Bragg diffraction $(k_d^{(2)})$ is in the opposite direction to the guided incident wave that establishes DFB. Symmetric (large radiative out-coupling) and antisymmetric (small radiative out-coupling) resonant modes are established at the edges of the photonic bandgap due to DFB. **c** A hybrid grating concept that combines second-order and fourth-order Bragg gratings. The second-order grating provides stronger DFB, and leads to establishing resonant optical modes with similar phase relationships to the grating as in **b**. An added fourth-order grating serves to enhance the out-coupling of the antisymmetric mode and reduce that of the symmetric mode. By adjusting design parameters, either of those modes could be excited in a DFB laser cavity that has a greater radiative efficiency compared to the excited mode for a second-order DFB cavity

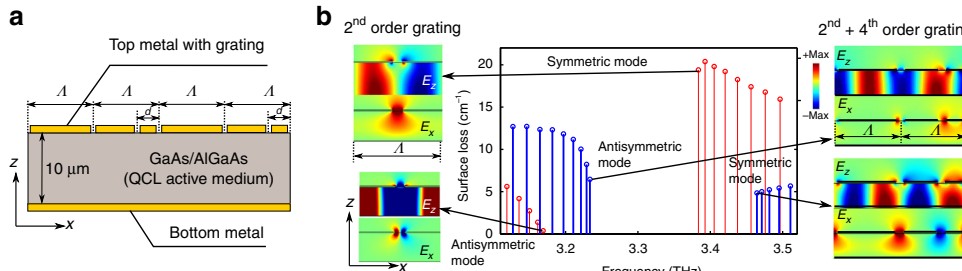

**Fig. 2** Comparison of hybrid DFB and second-order DFB for terahertz QCLs. **a** Illustration of a metallic cavity for terahertz QCLs in which slits are opened in the top metal cladding to implement a periodic grating[18]. A fourth-order grating is superimposed at an offset of length $d$ to the original second-order grating with periodicity $\Lambda$ to realize a hybrid grating structure as in Fig. 1c. **b** Mode-spectrum for a 1.4 mm long and infinitely wide cavity with DFB gratings ($\Lambda = 27$ μm, slit-width ~3 μm) computed with finite-element modeling method. Radiative surface-losses for various resonant modes for cavity with second-order gratings are plotted in red (thin-lines), and that for cavity with hybrid gratings ($d/\Lambda = 3/8$) in blue (thick-lines). The insets show electric field profiles for lower and upper band-edge modes respectively, of the photonic bandstructure for each type of gratings (the color-bar shown in this figure applies to all the plots of electric field profiles). The radiative loss is effectively determined by the amplitude and phase of the in-plane electric field ($E_x$) in the slits

($E_x$) field)[18]. In comparison, the cavity with the hybrid second-order and fourth-order gratings greatly enhances the surface-loss for the antisymmetric lower band-edge mode due to additional radiation from the slit corresponding to the fourth-order superposed grating. The loss for the symmetric upper band-edge mode is reduced from that of the second-order DFB structure. For the simulated case of $d/\Lambda$ in Fig. 2, the upper band-edge mode is of lower loss for the hybrid DFB grating, and will be excited in a lasing cavity. More importantly, the surface loss of the excited mode in laser cavity with hybrid DFB can be enhanced by an order of magnitude from that of the second-order DFB cavity. It may be noted that bandgap of the hybrid DFB cavity has a red-shift of ~0.08 THz in the shown simulation, which is due to the fact that a larger fraction of evanescent field propagates outside

the active medium that lowers the effective propagation index $n_{eff}$ of the guided waves.

For the symmetric mode excited in the case of hybrid DFB with a specific $d/\Lambda = 3/8$, $n_{eff}$ is close to ~3.2 according to Eq. (1), this relatively low $n_{eff}$ is due to the establishment of a strong surface plasmon polariton (SPP) field that propagates on the top of the active region as shown in Fig. 2b. In contrast, the antisymmetric mode has a larger $n_{eff} \sim 3.45$, which translates to a larger fraction of resonant mode confined inside the active medium.

The hybrid DFB grating offers flexibility in design by simply altering the offset $d$ at which the fourth-order grating is implemented as shown in the schematic in Fig. 2a. The relative radiative losses of the two band-edge modes could be modulated by adjusting $d/\Lambda$, which is useful to design for selective excitation

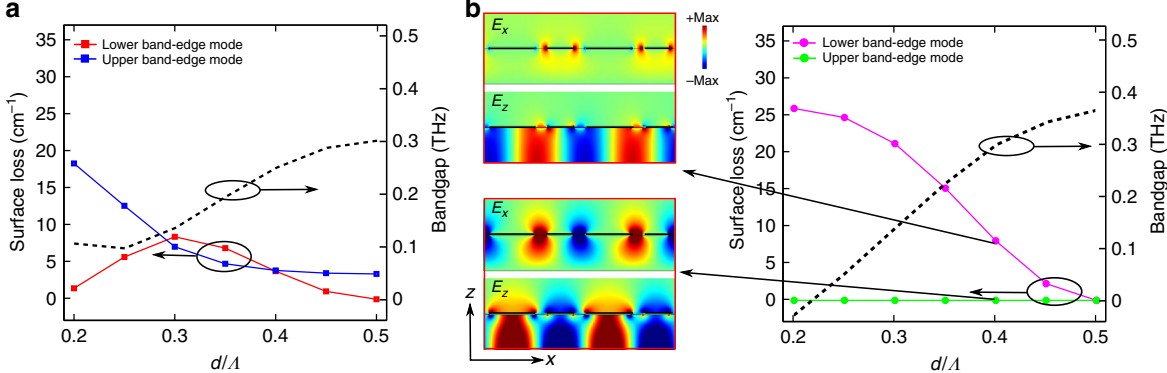

**Fig. 3** Design considerations and comparison with a dual-slit second-order DFB structure. **a** Computed surface loss of the band-edge modes for the hybrid DFB structure shown in Fig. 2a, plotted as a function of $d/\Lambda$. Aperture spacing $d$ is a design parameter that could be utilized to alter the respective losses and also the bandgap, which is also plotted. **b** Surface loss and bandgap is also plotted as a function of aperture spacing for a dual-slit second-order DFB structure (i.e., two slits are present in each grating period at the periodicity of $\Lambda$), which has been used for terahertz QCLs previously[26]. Electric field profiles near the center of cavities for both band-edge modes are plotted for the case of $d/\Lambda = 0.4$ as an example

of a lower or upper band-edge mode. This is shown in the plot of computed surface losses of the band-edge modes of the infinitely wide but finite-length terahertz QCL cavity in Fig. 3a, where simulation is done with same parameters for the grating as in Fig. 2. For a certain range of $d/\Lambda$ values, the loss of the upper band-edge mode could be lowered and hence such a mode could be selectively excited. For experimental results presented later, $d/\Lambda = 3/8$ was implemented to selectively excite an upper band-edge mode. Note that in this case, both band-edge modes lead to single-lobed beams in the far-field and symmetric/antisymmetric designations lose significance due to an overall constructive interference in the far-field for both band-edge modes (Supplementary Note 2 shows an intuitive explanation on single-lobed beam profile for both bang-edge modes), when considering the net radiation from every $2\Lambda$ length of the DFB structure. In this sense, the hybrid DFB structure could instead simply be called a fourth-order DFB scheme, but one that has a specific requirement for its unit cell of periodicity $2\Lambda$. We choose not to use this terminology since then it does not clearly indicate the operating mechanism of the DFB.

Given that the hybrid DFB for terahertz QCLs presented here is implemented with dual-slits in metal cladding in every alternate period of length $\Lambda$, it is intuitively compelling to consider whether the radiative loss for a second-order DFB for such QCLs could also be increased by introducing dual-slits in each period to avoid the nulls of the radiative field $E_x$ in one of the two slits. Such a DFB structure was indeed investigated experimentally in ref. [26]. However, it turns out, such a DFB structure also has an antisymmetric mode for one of its band-edge modes, which has negligible radiative coupling. In this case, the role of symmetry is reversed for the band-edge modes, and the upper band-edge mode now becomes the low-loss mode with destructive interference for radiative coupling into the far-field. The behavior of the surface-losses for the two band-edge modes and the representative electric field profiles for the dual-slit second-order grating structure are shown in Fig. 3b. This type of structure could potentially be used to increase output power if a large $d/\Lambda$ was chosen to make the photonic bandgap large, such that the upper band-edge modes lie outside the gain region of the active medium. This was indeed the strategy employed in ref. [26] to increase power output from SE DFB terahertz QCLs moderately. While the dual-slit structure may appear to offer somewhat similar functionality to the hybrid DFB structure, the latter is fundamentally different in its functionality

and effectiveness to improve radiative efficiency for surface-emission.

**Experimental results**. Experimental results from representative SE terahertz QCLs implemented with hybrid DFB gratings in pulsed mode of operation and mounted inside a Stirling cooler are shown in Fig. 4. The scanning electron microscope (SEM) image of the fabricated and mounted QCL chip in Fig. 4a shows several QCLs of varying dimensions located side by side. The results presented here are from QCLs of dimension 10×200 μm×1.5 mm. The choice of the length of cavity is made based on estimation of the DFB coupling strength, and is described in the Supplementary Note 1 and Supplementary Note 2, where the simulated energy-density profile along the length of the cavity for the chosen length is shown in Supplementary Figure 1. The hybrid DFB grating in the form of slits is implemented in the top metal cladding. Figure 4b shows the light–current (L–I) curves versus heat-sink temperature, current–voltage (I–V) curve at 62 K, and also spectra as a function of bias at 62 K. The QCL emits in single-mode at all bias conditions at ~3.39 THz, and operated up to maximum temperature of 105 K. Figure 4c shows the measured far-field radiation pattern, which is single-lobed and a characteristic of symmetric mode excitation for the resonant-mode of the DFB structure. The full-width half-maximum divergence is ~5° × 25° that closely matches the result from full-wave finite-element simulation of the DFB cavity as presented in Supplementary Figure 1. Finally, the robustness of the DFB scheme in exciting the desired mode based on lithographically defined periodicity is exemplified from the lasing spectra of three different QCLs with different $\Lambda$ shown in Fig. 4d. All QCLs showed single-mode operation in the entire dynamic range and the lasing frequencies scale with $\Lambda$ with an effective propagation index $n_{eff} \sim 3.16$ for the guided modes. This relatively low $n_{eff}$ certifies that the upper band-edge mode is excited for these QCLs as designed.

The primary contribution of this work is the high radiative efficiency of the hybrid DFB scheme. A peak optical power output of $170 \pm 3$ mW at 62 K was measured for the QCL reported in Fig. 4b, which is the power detected from the power meter without making any corrections for the imperfect collection efficiency and optical losses from the cryostat window. The wall-plug efficiency of this device is ~0.78% and a slope efficiency of $993 \pm 15$ mW A$^{-1}$ (differential quantum efficiency of 71 photons/

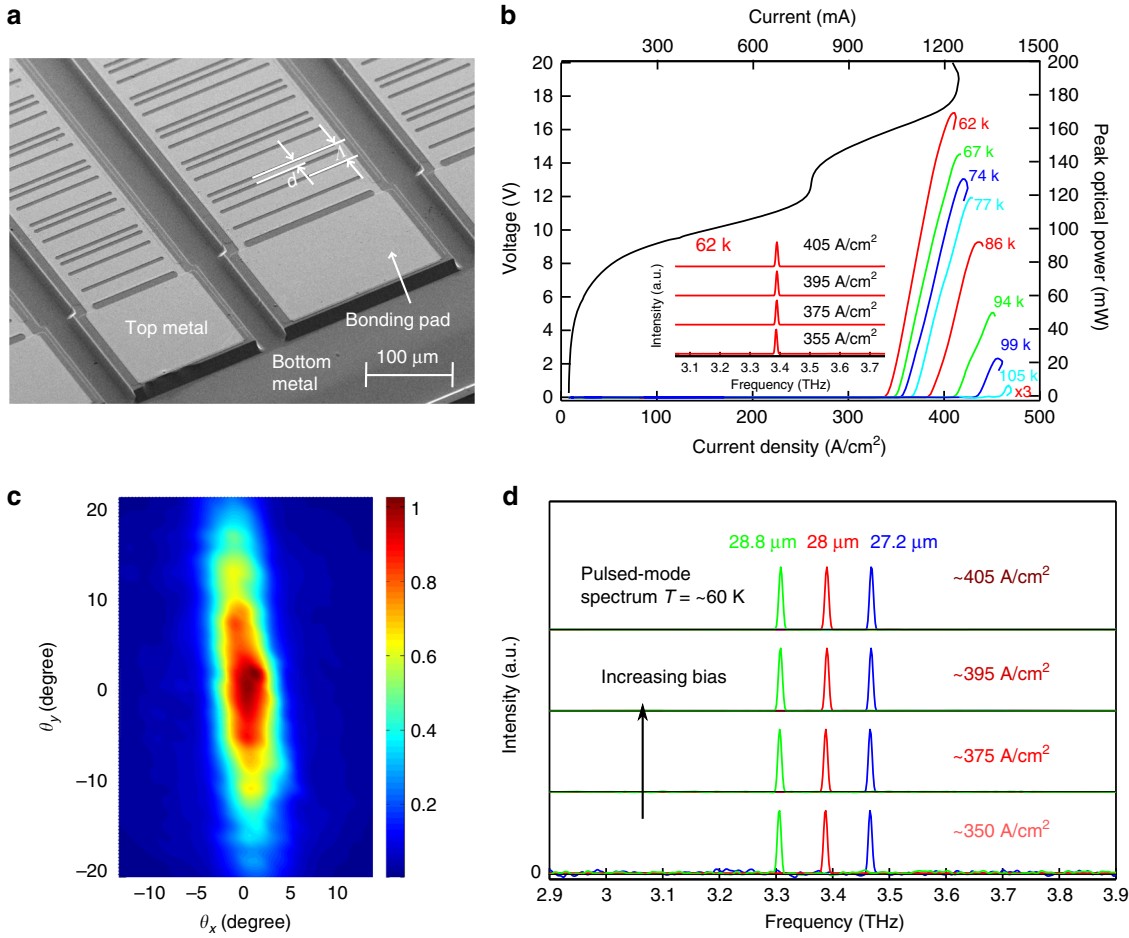

**Fig. 4** Experimental results. **a** Scanning electron microscope image of the fabricated terahertz QCLs with hybrid DFB gratings as in Fig. 2a. **b** Current–voltage (*I*–*V*), and spectral intensity (inset) at different electric bias at a heat-sink temperature of 62 K, and light–current characteristics at different temperatures measured in pulsed mode of operation. The terahertz QCL is of dimensions 10 μm×200 μm×1.5 mm, grating period $\Lambda = 28$ μm, and $d/\Lambda = 3/8$. **c** Far-field radiation pattern (optical intensity) measured at 62 K close to the peak-bias (~405 A cm$^{-2}$) at a distance of 40 mm from the QCL in the surface-normal direction. $\theta_x$ and $\theta_y$ are angles with respect to the surface normal along the longitudinal and lateral dimensions of the QCL cavity, respectively. **d** Spectral characteristics of three different QCLs located neighboring each other on the wafer with different grating periods, and $d/\Lambda = 3/8$ for each QCL

electron) is estimated from the slope of the 62 K *L–I* using linear curve fitting in the range of 20−80% of bias range of the QCL. The differential and slope-efficiencies are highest achieved to-date from any terahertz QCL including that from Fabry–Pérot QCLs with single-plasmon waveguides, which had demonstrated the best radiative efficiencies previously. For comparison, terahertz QCLs with conventional second-order gratings were also fabricated from the same MBE wafer. *L–I* data from one such representative QCL with similar cavity dimensions is shown in Supplementary Note 3, which achieved a peak power output of 50 mW, a maximum operating temperature of 129 K, a wall-plug efficiency of ~0.18% and a slope efficiency of ~80 mW A$^{-1}$ at 63 K. While the peak power outputs cannot be compared since the dynamic range for lasing is higher for the conventional second-order DFB device, the slope efficiency for the QCL with hybrid DFB is more than an order of magnitude higher compared to the conventional second-order DFB QCL that validates the strong enhancement of radiation due to the new DFB technique. However, it has to be noted that the comparison of power output of hybrid DFB with conventional second-order DFB is not truly reflective of improvement in out-coupling efficiency due to the fact that any variation of the boundary condition near the

ends of cavities would significantly influence the emission loss of conventional second-order DFB[18]. Therefore *L–I* data from a Fabry–Pérot QCL with a dimension of 10 μm×150 μm×1.5 mm is also shown in Supplementary Note 4 for comparison, which shows that the maximum operating temperature of this active medium is 137 K. The temperature performance from this active medium is relatively modest compared to state-of-the art terahertz QCLs, and hence, a significant increase in power performance is expected with better performing active medium that could achieve a greater dynamic range in lasing with high temperature performance. To futher reveal the high performance of this hybrid second-order and fourth-order DFB mechanism, a comparison of this scheme with graded photonic structure[9] based on numerical simulation is presented in Supplementary Note 5.

## Discussion
In conclusion, here we have described a new DFB method to significantly increase radiative efficiency for surface-emitting solid-state lasers based on the widely used second-order Bragg gratings. A hybrid second-order and fourth-order DFB grating is

shown as an effective method to excite a symmetric radiative mode with large radiative out-coupling in a single-lobed beam, which is also simpler to implement compared to existing modifications to second-order gratings in literature. The hybrid DFB scheme is implemented for terahertz QCLs to realize a single-mode QCL with highest reported optical power output (170 mW) to-date. A record-highest slope-efficiency (993 mW A$^{-1}$) and differential quantum efficiency (71 photons per electron) is also experimentally demonstrated when compared to all previously reported terahertz QCLs in literature (including that from multi-moded Fabry–Pérot QCLs). These values suggest that the radiative efficiency realized for the QCL is approximately one-third of the maximum theoretically possible in absence of any loss mechanisms. In principle, the hybrid DFB technique should also be applicable to semiconductor lasers at near-infrared and mid-infrared wavelengths to increase the power output for single-mode surface-emitting lasers at those wavelengths.

## Methods

**Finite-element modeling**. All of the simulations were carried out by using COMSOL Multiphysics 4.4. A module of Electromagnetic waves, Frequency Domain (ewfd) under the catalog of Optics was utilized to calculate the eigenmodes of various kinds of DFB laser structures shown in this paper. In order to obtain accurate information of the emission loss, The active region is modeled as lossless and the metal is modeled to be perfect electrical conductors, the highly doped contact layer serving as absorbing boundaries of the cavity is implemented using a complex dielectric constant computed using Drude-model and a perfect-matching layer for absorbing boundary echoes was adopted to wrap all the borders. The specific details of the modeling for both 2D and 3D simulations are same as that in ref. [27], in which case, the computed loss is the sum of loss at absorbing boundaries as well as that due to radiation (out-coupling). By analyzing the eigenfrequencies and their corresponding radiation losses, the lasing frequency as well as the far-field beam patterns can be estimated.

**Materials**. The active medium of the THz-QCLs is based on a three-well resonant-phonon design with GaAs/Al$_{0.15}$Ga$_{0.85}$As superlattice (design RT3W221YR16A, MBE wafer VB832, with a layer sequence of 57/18.5/ 31/9/28.5/16.5 (starting from the injector barrier) where the thicknesses are in monolayers (ML, 1 ML = 2.825 Å), and was grown by molecular beam epitaxy, with 221 cascaded periods, leading to an overall thickness of 10 μm. The design is similar to the three-well QCL designs in [28, 29] with minor modifications to achieve peak gain centered around a frequency of 3.3 THz. The QCL superlattice has an average $n$-doping of 5.7e15 cm$^{-3}$ and surrounded by 0.1 μm and 0.05 μm thick highly doped GaAs contact layers doped at 5e18 cm$^{-3}$ on either sides of the superlattice. A 200 nm thick Al$_{0.50}$Ga$_{0.50}$As layer was grown as an etch-stop layer preceding the entire stack.

**Device fabrication**. Cu–Cu based metallic waveguides were fabricated using standard thermo compression wafer-bonding technique. Following wafer-bonding and substrate removal, positive-resist lithography was used to selectively etch away the 0.1 μm thick highly doped GaAs layer from almost all locations where top-metal cladding would exist on individual cavities by H$_2$SO$_4$:H$_2$O$_2$:H$_2$O etchant in 1:8:80 concentration. A 10 μm wide highly doped GaAs layer below the top-metal cladding was left unetched at the regions close to both longitudinal and lateral facets, serving as the longitudinal and lateral absorbing boundary to ensure the excitation of the desired mode as the lowest-loss lasing mode as described in ref. [27]. A sequence of Ti/Cu/Au were deposited as top (20/200/100 nm) metallic layers, in which an image-reversal lithography was implemented to form metallic gratings. DFB ridge cavities then were processed by wet-etching using H$_2$SO$_4$:H$_2$O$_2$:H$_2$O etchant in 1:8:80 concentration. A Ti/Cu/Au (20/250/100 nm) contact was also used as the backside-metal contact for the finally fabricated QCL chips to assist in soldering. Before deposition of backside-metal of the wafer, the substrate was mechanically polished down to a thickness of 250 μm to improve heat-sinking.

**Experimental characterization**. During the light–current–voltage measurements, A pulse of 300 ns duration with 100 kHz signal cycle (3.0 % duty cycle) was chosen to drive the devices presented in this paper on a cold-stage of a Stirling-cooler (which is operating at ~62 K). Under the same conditions, the absolute power was calibrated using a thermopile power meter (model number: Scientech AC2500 with AC25H) as is reported without any corrections to the detected signal. No focusing

optics were used in this process except a high-density polyethylene window on the cryocooler. The reported spectra were measured using a Fourier-transform infrared-spectrometer (BRUKER; VERTEX 70 v) by operating the devices at 100 kHz with a 300 ns pulse duration (3.0 % duty cycle). Far-field beam patterns were measured with a pyroelectric detector mounted on a 2D motorized scanning stage, which was placed at 40 cm from the DFB lasers, with maximum scan angle ± 26.5° in both two directions. The devices was operated near the peak power operated at 100 kHz with a 300 ns pulse duration and electronically modulated with pulse-trains at 1000 Hz (1.5 % duty cycle).

**Data Availiability**. All relevant data related to numerical simulation, experimental results will be preserved with Sushil Kumar at Lehigh University, the data sets are accessible via the corresponding author.

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

## Acknowledgements

This work is supported by the U.S. National Science Foundation under Grants: ECCS 1351142, ECCS 1609168, and CMMI 1437168. It was performed, in part, at the Center for Integrated Nanotechnologies, an Office of Science User Facility operated for the U.S. Department of Energy (DOE) Office of Science. Sandia National Laboratories is a multimission laboratory managed and operated by National Technology and Engineering Solutions of Sandia, LLC., a wholly owned subsidiary of Honeywell International, Inc., for the U.S. Department of Energy's National Nuclear Security Administration under contract DE-NA-0003525.

## Author contributions

S.K. conceived the original idea of a hybrid grating and supervised the project. Y.J. further developed the idea through analysis and simulations, and techniques for the practical implementation. Y.J. and C.W. performed numerical simulations. L.G. and Y.J. fabricated the devices. Y.J., J.C., and L.G. developed the experimental setup and conducted the measurements. J.L.R. was responsible for growth of the QCL material by molecular beam epitaxy. S.K. and Y.J. wrote the manuscript with inputs from all other authors.

## Additional information

**Competing interests:** The authors declare no competing interests.

