## [Peer Review File(PDF 213 kb) · Nature Communications]

Reviewers' comments:

Reviewer #1 (Remarks to the Author):

I commend the authors for their thorough revisions of the manuscript, in response to my comments. This is indeed a truly novel concept for efficient extraction of radiation for THz QCL and thus is eminently suitable for publication in Nature Communications.

I have only one suggested change in nomenclature, and a couple of minor corrections which can be found in the attached file.

Reviewer #2 (Remarks to the Author):

Jin and authors have extensively responded to my previous comments. They have discussed in detail the comparison with reference 7 of original submission and the advances and novelty made in this new manuscript. As detailed in my previous review, the paper is well written, the data is very good, and a novel concept on a hybrid grating has been proposed and realized for high slope efficiencies from THz QCLs.

I also thank the authors for the in-depth discussion on the comparison between 2nd order DFBs and other surface emitting structures. This has highlighted the gain in proposing a hybrid grating structure, as well as the difficulty in comparing these structures. Nonetheless, the authors have extensively compared the 2nd order DFB with the hybrid structure in the first part of the paper through electromagnetic simulations, which makes it normal to compare the results from these structures. I would therefore suggest that the authors remark close to the simulation part of the paper their point regarding boundary conditions of 2nd order DFB and how this would affect the results. (This has been added to the experimental part but should be mentioned in the simulation section).

Regarding the simulation comparison between the GPH and hybrid structure in the supplementary material, although it seems that the electric field distribution is indeed more concentrated in the GPH structures, I suggest that the authors use a similar device lengths to show that this indeed the case.

Minor point. From the far-field, the authors state that the symmetric mode has been excited in the hybrid grating case (page 9). However, from the supplementary material, the antisymmetric mode would also be single lobed. Could the authors clarify, if possible, which mode is excited (both modes have similar surface loss). Would the frequency emission give an indication of which mode is excited, rather than the far-field?

Taking the authors response, I believe that the concepts in this work are novel, with further perspectives to enhance the output power from THz QCLs. Further, the impact could be wide-ranging where it can be applied to other spectral domains, impacting laser diodes as well as QCLs in terms of high power surface emission. I therefore believe that the paper is well-suited to Nature Communications.

Reviewer #3 (Remarks to the Author):

2nd Report on

„High power surface emitting THz laser with hybrid second and fourth-order Bragg grating“

By Jin et al.

The authors have answered my questions in detail and improved the manuscript. The manuscript is now suitable for Nature Communications. The authors should consider to include their data on the symmetric far field in this manucrypt. It would make the manuscript much stronger.

Reviewer#1

I commend the authors for their thorough revisions of the manuscript, in response to my comments. This is indeed a truly novel concept for efficient extraction of radiation for THz QCL and thus is eminently suitable for publication in Nature Communications.

I have only one suggested change in nomenclature, and a couple of minor corrections which can be found in the attached file.

The authors have revised the text in a highly satisfactory manner, in response to my comments; thus, the manuscript has become a lot more clearer. Given the absolutely novel type of surface-emitting (SE) distributed-feedback (DFB) THz quantum cascade laser that is proposed and demonstrated, and which achieves record-high single-mode peak output power and slope efficiency value for THz lasers, I strongly recommend publication in Nature Communications.

I have one minor suggested revision/correction. In my opinion, the factor in the slope efficiency that the authors call internal quantum efficiency is better described as internal differential efficiency [see Snowton and Blood, IEEE J. Sel. Topics Quantum Electron . 3, 491 (1997)] since it captures the differential nature of the carrier usage above threshold both as far as the injection efficiency (i.e., tunneling injection efficiency + % of carriers not lost due to inelastic and/or elastic scattering) and as far as the lasing transition itself (so-called differential transition efficiency in QCLs). Thus, I suggest replacing “internal quantum efficiency” with “internal differential efficiency” or simply with “internal efficiency”, and insert in the last sentence of section G (in Supplementary Information) “, for a given internal-efficiency value,” between “slope-efficiency” and “ is an intuitive indicator”.

Minor corrections/additions:

Main text:

Page 5, line 12:

Replace “hence” with “turn”

lines 19-22: The sentence starting with “For...” should be rephrased for clarity. Although the authors refer, in their response letter, to a comparison between the wall-plug efficiency of the novel device and that for conventional 2nd -order DFB device in the revised manuscript, they seem to have forgotten to include it in the text. Also what is the value for the new device: 0.77% or 0.78 % at 62K ?

Supplementary Information

In Eq. 3, the subscripts for α in the numerator and the first term in the denominator are not the same as in Eq. 2. Please correct accordingly.

Response:

Based on the reviewer's suggestions, we have made all the corrections mentioned above. We also replaced “internal quantum efficiency” by “internal differential efficiency” in the manuscript.

In the revised manuscript as well as supplementary material, we mentioned that the wall-plug efficiency of the hybrid DFB is $\sim 0.78\%$ and the wall-plug efficiency of conventional 2nd order DFB with similar dimensions of the hybrid DFB is $\sim 0.18\%$. (Page. 9 and Page. 10 of the main paper, Section. C of the supplementary materials)

Reviewer#2

Jin and authors have extensively responded to my previous comments. They have discussed in detail the comparison with reference 7 of original submission and the advances and novelty made in this new manuscript. As detailed in my previous review, the paper is well written, the data is very good, and a novel concept on a hybrid grating has been proposed and realized for high slope efficiencies from THz QCLs.

(a). I also thank the authors for the in-depth discussion on the comparison between 2nd order DFBs and other surface emitting structures. This has highlighted the gain in proposing a hybrid grating structure, as well as the difficulty in comparing these structures. Nonetheless, the authors have extensively compared the 2nd order DFB with the hybrid structure in the first part of the paper through electromagnetic simulations, which makes it normal to compare the results from these structures. I would therefore suggest that the authors remark close to the simulation part of the paper their point regarding boundary conditions of 2nd order DFB and how this would affect the results. (This has been added to the experimental part but should be mentioned in the simulation section).

(b). Regarding the simulation comparison between the GPH and hybrid structure in the supplementary material, although it seems that the electric field distribution is indeed more concentrated in the GPH structures, I suggest that the authors use a similar device lengths to show that this indeed the case.

(c). Minor point. From the far-field, the authors state that the symmetric mode has been excited in the hybrid grating case (page 9). However, from the supplementary material, the antisymmetric mode would also be single lobed. Could the authors clarify, if possible, which mode is excited (both modes have similar surface loss). Would the frequency emission give an indication of which mode is excited, rather than the far-field

Taking the authors response, I believe that the concepts in this work are novel, with further perspectives to enhance the output power from THz QCLs. Further, the impact could be wide-ranging where it can be applied to other spectral domains, impacting laser diodes as well as QCLs in terms of high power surface emission. I therefore believe that the paper is well-suited to Nature Communications.

Response:

We thank the reviewer for the detailed comments and valuable suggestion. As the reviewer indicated, an exact quantitative comparison of hybrid DFB grating in our manuscript with the GPH or other high-performance surface-emission grating for terahertz QCLs as reported in prior literature is difficult.

(a). To the point about comparison with conventional 2nd order DFB gratings, we would like to mention that the discussion centered around Fig. 2 of the paper is primarily to highlight the electric-field distribution of the band-edge modes, which directly relates to the intensity of out-coupled radiation. In this regard, boundary conditions at the end-facets of the cavity have a minor role to play. For robust single-mode operation, it is desirable to have absorbing boundaries to eliminate reflections from the facet. For relatively long cavities ($>2\text{mm}$ long), our experience suggests the boundary conditions do not impact surface-loss even for conventional 2nd order DFB grating terahertz QCLs. For shorter cavities, the affect of boundary condition on surface-loss was investigated in Ref. 18 (Kumar et al., Opt. Express, 2007). A statement to this effect has now been added in main manuscript (2nd paragraph, Page. 5) as

“The simulations are shown for cavities with absorbing boundaries implemented at the longitudinal ends of the cavities, which minimizes reflections from the end-facets and leads to robust single-mode operation in the DFB cavity. However, precisely implemented high-reflectivity end-facets could also potentially be used for conventional second-order DFB cavities to enhance the radiative loss in such cavities further [18].”

Another statement about the effect of boundary conditions on conventional second-order DFB has been added to **section C** of the supplementary materials.

(b).

To the point of the comparison of simulation results between the GPH and hybrid DFB structures, the cavity length we used for the GPH grating structure is as same as that presented in Ref. 9. In other words, Fig. S5 presents the electric energy density distribution of the actual lasing devices with GPH and hybrid DFB schemes respectively, which we thought would be a more valuable comparison for the reader. On the other side, we believe that the total length of cavity with metallic gratings ($\sim 700\mu\text{m}$) for

the GPH design was not arbitrarily determined. The length of the cavity was likely chosen after careful numerical calculations similar to that for our structure as presented in **Section A** of the supplementary material. That is to say, any attempt on making the length of GPH laser cavities longer or shorter is less likely to increase the power output from a single cavity.

To further investigate reviewer's concern about this issue, we have done additional simulations for the cavities with GPH structures with various lengths of cavity. As shown in the above image, when a similar device length for GPH structure with that of hybrid DFB scheme is chosen, the electric field distribution is still highly concentrated in the center of laser cavity with a FWHM $\sim 180 \mu\text{m}$ (that is the same as that for the much shorter cavity). The large non-uniformity of electric field distribution of GPH is not determined by the length of laser cavity but is due to specific distributed-feedback coupling of the GPH gratings itself. Although we have not included this additional simulation result in revised submission, we have mentioned a statement to this effect in section E of the supplementary information, as

“Simulation of further increase in length of cavities of GPH shows a similar electric-field distribution with similar FWHM of electric energy density profile, which verifies that the large non-uniformity of electric field distribution of GPH is not determined by the length of laser cavities but is due to its specific DFB coupling of the GPH itself.”

(c). The reviewer is correct in identifying that both "symmetric" and "anti-symmetric" modes would show a similar single-lobed far field beam pattern for our hybrid DFB structure. Hence, indeed the frequency of emission gives the indication about which mode is excited. Alternately, the effective mode index of the excited mode also indicates which band-edge mode is the excited mode. As per Fig. 2 of the main manuscript, for a given grating period $\Lambda = 27 \mu\text{m}$ and $d/\Lambda = 3/8$, the symmetric mode would be excited as the lowest loss mode, in which case, the effective mode index n_{eff} of the resonant mode is ~ 3.2 according to Eq.(1). This relatively low n_{eff} is due to the establishment of a strong SPP field that propagates on the top of the active region as shown in Fig.2 (b). In contrast, the anti-symmetric mode has a larger $n_{\text{eff}} \sim 3.45$, which translates to a larger fraction of resonant-mode confined inside the active medium. Therefore, n_{eff} could be treated as a good indicator for clarifying which band-edge mode is excited. We have now added a statement to this effect in the main manuscript (Page. 6),

"For the symmetric mode preferred to be excited in the case of hybrid DFB with a specific $d/\Lambda = 3/8$, n_{eff} is close to ~ 3.2 according to equation (1), this relatively low n_{eff} is due to the establishment of a strong surface plasmon polariton (SPP) field that propagates on the top of the active region as shown in Fig. 2 (b). In contrast, the anti-symmetric mode has a larger $n_{\text{eff}} \sim 3.45$, which translates to a larger fraction of resonant mode confined inside the active medium."

Reviewer#3

2nd Report on “High power surface emitting THz laser with hybrid second and fourth-order Bragg grating” By Jin et al.

The authors have answered my questions in detail and improved the manuscript. The manuscript is now suitable for Nature Communications. The authors should consider to include their data on the symmetric far field in this manuscript. It would make the manuscript much stronger.

Response:

We thank the reviewer for the suggestion about including our data on the symmetric far field in this manuscript. The symmetric far-field was obtained after designing a custom THz lens. The results could

not be presented in this manuscript in a concise form. The experimental investigation for that result is detailed and included some new concepts, and is currently being prepared for submission as a separate manuscript to another journal.

With this letter and our revised submission, we hope to have addressed all the reviewers' concerns. We thank the reviewers again for their valuable time and effort in reviewing our manuscript, and for their constructive criticisms.

Sincerely,

Yuan Jin

(on behalf of all the authors)

REVIEWERS' COMMENTS:

Reviewer #2 (Remarks to the Author):

The authors have responded extensively to my second set of comments and the manuscript is well adapted to Nature Comm